# Ocean–Continent Conversion in Beishan Orogenic Belt: Evidence from Geochemical and Zircon U-Pb-Hf Isotopic Data of Luotuoquan A-Type Granite

**Wenliang Chen [1], Minjie Zhang [2,3,*], Guanghuo Tao [1], Xiaofeng Li [1], Qian Yu [2,3], Xiaojie Fan [2,3] and Jingwei Zhang [2,3]**

[1] Hebei Institute of Geological Survey, Shijiazhuang 050000, China; hebeiddycwl@163.com (W.C.); hebeiddytgh@163.com (G.T.); hebeiddylxf@163.com (X.L.)
[2] Hebei Key Laboratory of Strategic Critical Mineral Resources, Hebei GEO University, Shijiazhuang 050031, China; yuqian_hgu@163.com (Q.Y.); fxj18233385502@163.com (X.F.); zhangjinglv@163.com (J.Z.)
[3] College of Resources, Hebei GEO University, Shijiazhuang 050031, China
* Correspondence: mj-zhang@hgu.edu.cn

**Abstract:** Devonian magmatism is one of the most important tectonothermal events in the Central Asian Orogenic Belt (CAOB). However, little is known regarding the petrogenesis and geodynamic setting of the widely distributed Devonian granitoids in the eastern Southern Beishan Orogenic Belt (SBOB). Early-Devonian granitic magmatism has been recognized from the Luotuoquan area, and the granites were emplaced between 404.9 Ma and 399.4 Ma according to LA-ICPMS zircon U–Pb dating. Geochemically, the granites exhibit high $SiO_2$ and $Al_2O_3$ contents and are enriched in light rare earth elements as well as Rb, Th, Nd, Zr, and Hf, while being depleted in heavy rare earth elements and Ba, U, Sr, and Ti, with distinct rare earth element fractionation and pronounced negative Eu anomalies. According to the comprehensive analysis, they closely resemble the features typically associated with A-type granites. The zircons $\varepsilon_{Hf}$(t) values are within the range of +0.90–+5.19 (averaged 3.23) for the monzogranite and syenogranite, whereas their $T_{DM2}$ values fall between 1.05 and 1.34 Ga, suggesting that the magma source of the monzogranite–syenogranite originated from the partial melting of the Mesoproterozoic crust dominated by metagreywackes. Furthermore, the monzogranite and syenogranite exhibit high temperatures (average 782 °C), thin crustal thickness (average 30 km), and A-type characteristics, suggesting their formation in post-collision extensional settings. We propose the closure of the Beishan Ocean occurred before the early Devonian, followed by a transition in the Southern Beishan Orogenic Belt from a compressional to an extensional setting.

**Keywords:** zircon U–Pb dating; geochemistry; A-type granite; Luotuoquan complex; Beishan Orogenic Belt

## 1. Introduction

The Central Asian Orogenic Belt (CAOB) is formed by the sequential accretion of multiple microcontinents, island arcs, seamounts, oceanic plateaus, and accretionary complexes from the early Neoproterozoic to the late Paleozoic [1–3]. The Beishan Orogenic Belt (BOB) connects the Tarim, Kazakhstan, and North China cratons [4–6] (Figure 1a) and is considered a critical southern Central Asian Orogenic Belt segment, vital for understanding crustal growth and tectonic evolution [5–16]. The Southern Beishan Orogenic Belt (SBOB), primarily consisting of the Shuangyingshan and Huaniushan units (Figure 1), is thought to record the collision and subduction among ancient microcontinents with Mesoproterozoic to Neoproterozoic basements [17–21]. However, the tectonic and thermal evolution of the SBOB remains unclear, particularly the timing of the Beishan Ocean closure during the early Paleozoic. Current views suggest the closure occurred in the later Silurian–early Devonian [22–25], Carboniferous [25], early Permian [26], or after the late Permian [27–29]. Thus, the Paleozoic crustal growth and geodynamics of the SBOB are rather controversial

compared to those of the Mesozoic [11]. Further investigation of the tectonic and crustal development of the SBOB during the early–late Paleozoic is needed.

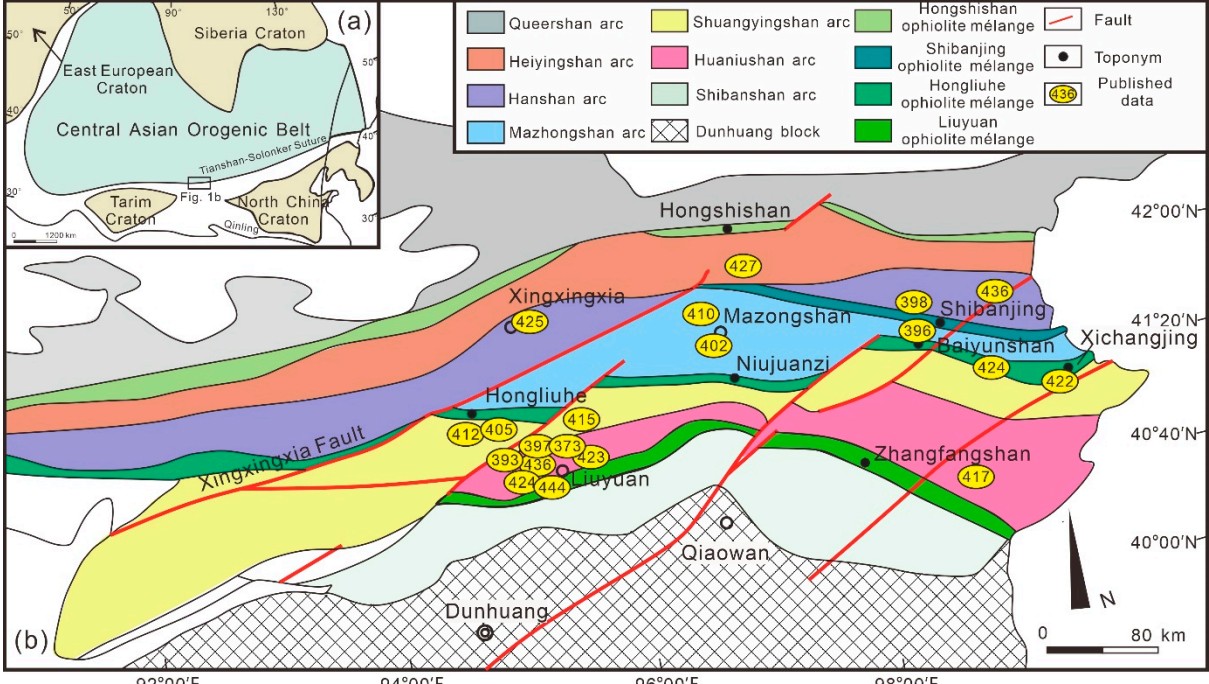

**Figure 1.** (**a**) Tectonic setting of the BOB (modified after [5]). (**b**) Simplified tectonic map of the Beishan orogenic collage and its adjacent area showing the tectonic subdivisions (modified after [5,13]). The zircon U-Pb age data of granitoids in (**b**) are from previous studies [4,6,10,30–35].

Herein, we present comprehensive whole-rock geochemical, in situ zircon U–Pb geochronology, and Hf isotopic data of A-type granites from the Luotuoquan complex. In addition to previously published findings, we discuss the genetic mechanism and tectonic settings of the early Paleozoic granitoids of the SBOB, elucidating their significant implications for understanding the tectonic evolution during this period in the BOB while also providing valuable insights into Beishan Ocean closure.

## 2. Geological Setting

The BOB comprises a complex assemblage of blocks, magmatic arcs, and ophiolitic mélanges that were formed through the subduction–accretion process of the Paleo–Asian Ocean [5]. Based on the spatial and temporal distributions of the ophiolitic mélanges and rock associations, the BOB is divided into several arcs [5], comprising (from north to south) the Queershan, Heiyingshan, Hanshan, Mazongshan, Shuangyingshan, Huaniushan, and Shibanshan arcs, which are separated by the Hongshishan, Shibanjing, Hongliuhe, and Liuyuan ophiolitic mélanges, respectively (Figure 1b).

The SBOB is situated within the Hongliuhe and Liuyuan ophiolitic mélanges [10] (Figure 1b) and comprises the Shuangyingshan and Huaniushan Units. In addition, the southern Beishan has recorded the subduction and collision of several microcontinental fragments and is characterized by early Paleozoic volcanic–sedimentary formations and intrusive plutons generated during early Paleozoic subduction [20,22,36,37]. Recent geochronological studies in the region reveal three stages of pluton emplacement in the Mid-Ordovician to late Silurian [6,23,38–40], early Devonian [4,6,40–42], and Late Devonian to early Carboniferous [6,38].

The Luotuoquan complex is situated in the eastern segment of the SBOB (Figure 1b) and intrudes into Silurian granitoids (Figure 2). It primarily consists of monzogranite, syenogranite, and minor basic rock. Field observations reveal that the syenogranite exhibits

weak mylonitization and has intruded into the monzogranite (Figure 2). The Devonian samples are derived from the Luotuoquan granite complex, comprising five monzogranite samples (PM03-2, PM03-26, PM03-41, PM12-20, and PM17-4 and three syenogranite samples (D5195, D8373, and PM06-34). The monzogranite samples display medium- to coarse-grained massive rocks, primarily composed of quartz (20–25 vol%), K-feldspar (35–40 vol%), plagioclase (30–35 vol%), and orientated biotite and/or muscovite (5 vol%) (Figure 3a,b). The syenogranite samples exhibit well-developed mylonitization and a granoblastic texture. They predominantly consist of K-feldspar (55–60 vol%), plagioclase (10–15 vol%, An = 5–10), and quartz (20–25 vol%) with minor biotite (5 vol%) (Figure 3c,d).

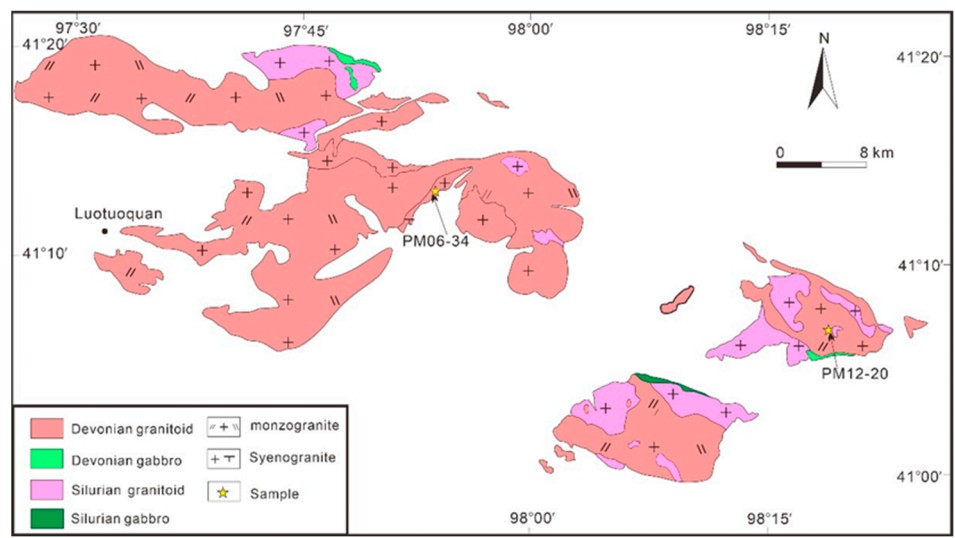

**Figure 2.** Geological map of the mafic–ultramafic intrusions and granite plutons in the Luotuoquan area.

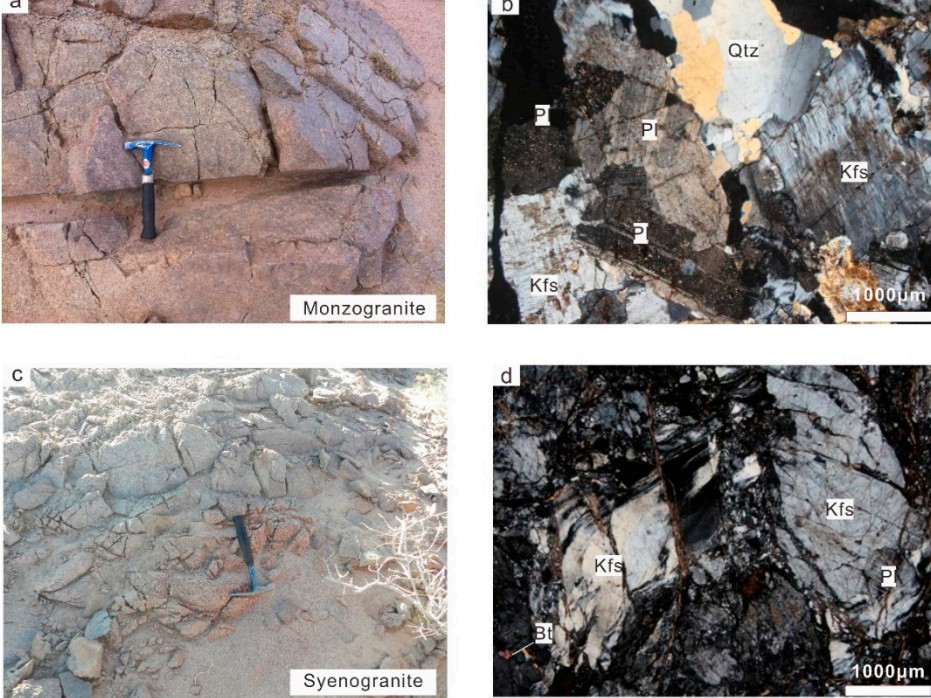

**Figure 3.** Field and microscopic photos of the Luotuoquan monzogranite (**a**,**b**) and syenogranite (**c**,**d**). Qtz-quartz; Pl-plagioclase; Kfs-potassium feldspar; Bt-biotite.

## 3. Analytic Methods

### 3.1. Zircon Dating and CL Imaging

Zircons were separated from the granite samples PM06-34 and PM12-20 of the Luo-tuoquan complex for laser ablation inductively coupled plasma mass spectrometry (LA-ICP-MS) U–Pb dating. Zircon grains were extracted using standard density and magnetic separation techniques. The selected zircon grains were handpicked under a stereoscopic microscope and mounted in epoxy resin before being polished to dissect the crystals in half for analysis. Cathodoluminescence (CL) and reflected-light photomicrographic analysis of the prepared sample targets were utilized to image the morphology and internal structure of the zircons to aid in selecting zircon grains for U–Pb dating. Zircon U–Pb dating analyses were conducted on a quadrupole inductively coupled plasma mass spectrometer (ICP-MS) (THERMO-ICAPRQ) coupled to a 193-nm ArF Excimer laser (Resolution-LR, Applied Spectra, West Sacramento, CA, USA) at Hebei Key Laboratory of Strategic Critical Mineral Resources. The laser spot size was set to 29 μm, the laser energy density was 3 J/cm$^2$, and the repetition rate was 8 Hz. Each analysis comprised a 10 s blank, a 40 s sampling ablation, and a 20 s sample-chamber flushing after the ablation. The ablated material was carried into the ICP-MS by the high-purity helium gas stream with a flux of 0.4 L/min. The whole laser path was fluxed with argon (0.9 L/min) to increase energy stability. A zircon 91,500 standard was used for external age calibration, and a zircon GJ–1 standard was used as a secondary standard to supervise the deviation of age calculation. Calibrations for trace element concentration were carried out using NIST SRM610 as an external standard and Si as the internal standard. ICPMSDataCal (Ver. 4.6) [43] and Isoplot 3.0 [44] programs were used for data reduction.

### 3.2. In Situ Lu-Hf Isotopes

In situ zircon Lu–Hf isotopic analyses were performed using a Neptune Plus MC–ICP–MS (Thermo Fisher Scientific, Brunswick, Germany) equipped with a Geolas 2005 excimer ArF laser ablation system (LambdaPhysik, Göttingen, Germany). All data on zircon were acquired in a single-spot ablation mode at a spot size of 44 μm. The energy density of laser ablation used in this study was ~7.0 J/cm$^{-2}$. Each measurement consisted of a 20-s acquisition of the background signal, followed by a 50-s acquisition of ablation signals. Instrumental conditions and data acquisition were as described by Wu et al. (2006) [45]. Zircon 91,500 was used as the reference standard. The chondritic ratios of $^{176}Hf/^{177}Hf = 0.282772$ and $^{176}Lu/^{177}Hf = 0.0332$ were used in our calculation of $\varepsilon_{Hf}(t)$ values [46]. Single-stage model ages (T$_{DM1}$) were calculated by reference to depleted mantle with a present day $^{176}Hf/^{177}Hf$ ratio of 0.28325 and $^{176}Lu/^{177}Hf$ ratio of 0.0384 [47]. The two-stage Hf model age (T$_{DM2}$), also interpreted as crust formation age, was calculated by projecting the zircon $^{176}Hf/^{177}Hf$ (t) back to the depleted mantle model growth curve, assuming a mean crustal value for Lu/Hf ($^{176}Lu/^{177}Hf = 0.015$) [48].

### 3.3. Whole-Rock Geochemical Analysis

Whole-rock geochemical analyses were performed at the Institute of Regional Geology Survey of Hebei Province. Fresh chips of whole-rock samples were powdered to 200 mesh using a tungsten carbide ball mill. Major and trace elements were analyzed by X-ray fluorescence (Axios X; PANalytical B.V.) and inductively coupled plasma mass spectrometry (XSeries II; Thermo Fisher Scientific), respectively. The analytical precision is generally better than 2% for major elements. For trace element analyses, sample powders were digested using HF + HNO$_3$ mixture in high-pressure Teflon bombs at 190 °C for 48 h or longer. The analytical precision is generally better than 5% for trace elements.

## 4. Result

### 4.1. Zircon U-Pb Dating and Lu-Hf Isotope Compositions

The zircons from monzogranite (PM12-20) and syenogranite (PM06-34) are similar in crystal morphology with sizes ranging from 80 to 200 μm and aspects ratios of 2:1–3:1.

The CL images reveal oscillatory zoning and rare inherited cores (Figure 4). The Th/U ratios are >0.1, indicating magmatic origin (Table 1). Sixteen analyses from sample PM12-34 (monzogranite) were concordant and yielded a weighted mean age of 404.9 ± 1.4 Ma (Figure 4a). Sixteen spots were analyzed for dating from sample PM06-12, apart from six spots (RZ7, RZ9-11, RZ13, and RZ15), which represent the age of the inherited core, the remaining ten analyses yielded a weighted mean age of 399.4 ± 5.1 Ma (Figure 4b).

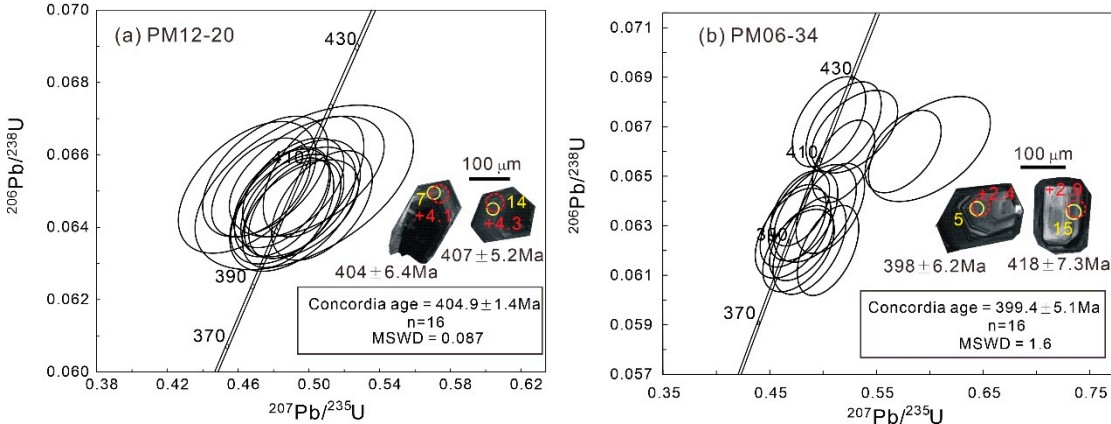

**Figure 4.** U–Pb concordia diagrams of zircons from Luotuoquan monzogranite (**a**) and syenogranite (**b**). The yellow line circle represents the spot of LA-ICP-MS analysis for U–Pb dating. The red dashed line circle represents the spot of LA-MC-ICP-MS analysis for Lu–Hf isotope compositions.

The analytical results are presented in Table 2, including $\varepsilon_{Hf}(t)$ values and model ages calculated using $^{206}Pb/^{238}U$ ages. The 32 spots from the Luotuoquan granites exhibit initial $^{176}Hf/^{177}Hf$ ratios ranging from 0.282205 to 0.282415 and positive $\varepsilon_{Hf}(t)$ values of +0.90–+5.19. Furthermore, the $T_{DM2}$ model ages range from 1.05 to 1.34 Ga.

### 4.2. Major and Trace Elements

The whole-rock major and trace element compositions of the five monzogranite samples and three syenogranite samples were analyzed, and the results are presented in Table 3.

#### 4.2.1. Major Elements

The samples possess high $SiO_2$ (71.04–76.00 wt.%), $K_2O$ (4.02–5.59 wt.%), and $K_2O + Na_2O$ (7.04–8.62 wt.%); however, they possess low $Al_2O_3$ (12.10–14.30 wt.%), MgO (0.07–0.63 wt.%), and CaO (0.76–2.53 wt.%). In the $SiO_2$ versus $K_2O + Na_2O$ diagram (Figure 5a), the samples are plotted within the granitoid fields, which is consistent with the petrographic observations. They further display low A/CNK values (molar $Al_2O_3/(CaO + Na_2O + K_2O)$) ranging from 0.85 to 1.08 and exhibit peraluminous and high K characteristics (Figure 5).

#### 4.2.2. Trace Elements

In the chondrite-normalized rare earth element patterns, the samples exhibit relative enrichment of light rare earth elements (LREE)($[La/Yb]_N$ = 2.63–16.87) and significant negative Eu anomalies (Eu/Eu* = 0.23–0.45), while the REE abundances range from 161.2 to 459.7 ppm (Figure 6a). All samples are depleted in Ba, U, Sr, P, and Ti and enriched in Rb, Th, Nd, Zr, and Hf relative to the primitive mantle (Figure 6b).

**Table 1.** LA-ICP-MS zircon U–Pb dating results of the Luotuoquan granites.

| No | Pb | U | Isotopic Ratios | | | | | | Isotopic Age (Ma) | | | | | |
|---|---|---|---|---|---|---|---|---|---|---|---|---|---|---|
| | (ppm) | | $^{207}$Pb/$^{206}$Pb | 1σ | $^{207}$Pb/$^{235}$U | 1σ | $^{206}$Pb/$^{238}$U | 1σ | $^{207}$Pb/$^{206}$Pb | 1σ | $^{207}$Pb/$^{235}$U | 1σ | $^{206}$Pb/$^{238}$U | 1σ |
| PM12-20: monzogranite | | | | | | | | | | | | | | |
| RZ1 | 75 | 294 | 0.0529 | 0.0028 | 0.4754 | 0.0234 | 0.0654 | 0.0011 | 324 | 120 | 395 | 16.1 | 409 | 7.2 |
| RZ2 | 101 | 457 | 0.0556 | 0.0026 | 0.4958 | 0.0213 | 0.0645 | 0.0009 | 439 | 104 | 409 | 14.5 | 403 | 6.1 |
| RZ3 | 84 | 420 | 0.0539 | 0.0020 | 0.4851 | 0.0179 | 0.0647 | 0.0010 | 365 | 88 | 402 | 12.3 | 404 | 6.5 |
| RZ4 | 69 | 360 | 0.0540 | 0.0021 | 0.4845 | 0.0178 | 0.0648 | 0.0009 | 372 | 87 | 401 | 12.2 | 405 | 6.3 |
| RZ5 | 120 | 529 | 0.0554 | 0.0020 | 0.5026 | 0.0188 | 0.0654 | 0.0011 | 428 | 81.5 | 413 | 12.7 | 408 | 6.6 |
| RZ6 | 72 | 296 | 0.0551 | 0.0028 | 0.4922 | 0.0244 | 0.0646 | 0.0012 | 417 | 115 | 406 | 16.6 | 403 | 7.1 |
| RZ7 | 74 | 300 | 0.0557 | 0.0026 | 0.4939 | 0.0214 | 0.0647 | 0.0010 | 443 | 99.1 | 408 | 14.6 | 404 | 6.4 |
| RZ8 | 110 | 486 | 0.0549 | 0.0020 | 0.492 | 0.0172 | 0.0645 | 0.0009 | 406 | 81.5 | 406 | 11.7 | 403 | 6.2 |
| RZ9 | 60.6 | 330 | 0.0510 | 0.0028 | 0.4651 | 0.0257 | 0.0651 | 0.0012 | 239 | 130 | 388 | 17.8 | 406 | 7.1 |
| RZ10 | 233 | 900 | 0.0559 | 0.0022 | 0.5074 | 0.0188 | 0.0646 | 0.0010 | 456 | 82.4 | 417 | 12.7 | 403 | 6.3 |
| RZ11 | 56.3 | 231 | 0.0548 | 0.0026 | 0.4878 | 0.0211 | 0.0645 | 0.0010 | 467 | 106 | 403 | 14.4 | 403 | 6.4 |
| RZ12 | 34.4 | 149 | 0.0551 | 0.0048 | 0.4974 | 0.0408 | 0.0651 | 0.0015 | 413 | 193 | 410 | 27.7 | 406 | 9.4 |
| RZ13 | 127 | 530 | 0.0544 | 0.0019 | 0.4882 | 0.0171 | 0.0644 | 0.0010 | 387 | 81.5 | 404 | 11.7 | 402 | 6.8 |
| RZ14 | 86 | 388 | 0.0545 | 0.0022 | 0.4912 | 0.0189 | 0.0651 | 0.0009 | 391 | 90.7 | 406 | 12.8 | 407 | 5.2 |
| RZ15 | 76 | 326 | 0.0550 | 0.0032 | 0.4988 | 0.0289 | 0.0654 | 0.0011 | 413 | 131 | 411 | 19.6 | 408 | 7.3 |
| RZ16 | 82 | 360 | 0.0524 | 0.0021 | 0.4729 | 0.0184 | 0.0651 | 0.0009 | 306 | 92.6 | 393 | 12.7 | 407 | 6.1 |
| PM06-34: syenogranite | | | | | | | | | | | | | | |
| RZ1 | 48.3 | 213 | 0.0554 | 0.0025 | 0.4770 | 0.0221 | 0.062 | 0.0011 | 428 | 104 | 396 | 15.2 | 387 | 6.4 |
| RZ2 | 51.2 | 227 | 0.0549 | 0.0029 | 0.4719 | 0.0234 | 0.0627 | 0.0012 | 406 | 117 | 392 | 16.1 | 392 | 7.3 |
| RZ3 | 46.1 | 192 | 0.0567 | 0.0031 | 0.4925 | 0.0255 | 0.0635 | 0.0013 | 480 | 114 | 407 | 17.4 | 397 | 8.1 |
| RZ4 | 114 | 422 | 0.0546 | 0.0022 | 0.4798 | 0.0198 | 0.0623 | 0.0010 | 394 | 90.7 | 398 | 13.6 | 390 | 6.1 |
| RZ5 | 123 | 585 | 0.0537 | 0.0021 | 0.4785 | 0.0182 | 0.0637 | 0.0010 | 367 | 88.9 | 397 | 12.5 | 398 | 6.2 |
| RZ6 | 92 | 305 | 0.0574 | 0.0026 | 0.5068 | 0.0212 | 0.0637 | 0.0012 | 506 | 100 | 416 | 14.3 | 398 | 7.2 |
| RZ7 | 77 | 337 | 0.0565 | 0.0028 | 0.5297 | 0.0266 | 0.0668 | 0.0011 | 472 | 105 | 432 | 17.7 | 417 | 7.0 |
| RZ8 | 61.6 | 384 | 0.0585 | 0.0022 | 0.5046 | 0.0190 | 0.0620 | 0.0012 | 550 | 83.3 | 415 | 12.8 | 388 | 7.1 |
| RZ9 | 85 | 398 | 0.0623 | 0.0026 | 0.5750 | 0.0235 | 0.0660 | 0.0011 | 687 | 88.9 | 461 | 15.1 | 412 | 7.2 |
| RZ10 | 98 | 447 | 0.0570 | 0.0022 | 0.5192 | 0.0193 | 0.0656 | 0.0011 | 500 | 83.3 | 425 | 12.9 | 410 | 6.1 |
| RZ11 | 54.2 | 278 | 0.0547 | 0.0025 | 0.5056 | 0.0227 | 0.0672 | 0.0012 | 398 | 99 | 415 | 15.3 | 419 | 7.5 |
| RZ12 | 52.6 | 241 | 0.0547 | 0.0025 | 0.4720 | 0.0204 | 0.0630 | 0.0011 | 398 | 102 | 393 | 14.1 | 394 | 7.2 |
| RZ13 | 39.9 | 168 | 0.0654 | 0.0038 | 0.5996 | 0.0377 | 0.0661 | 0.0014 | 787 | 122 | 477 | 23.9 | 412 | 8.1 |
| RZ14 | 75 | 267 | 0.0557 | 0.0027 | 0.4814 | 0.0214 | 0.0634 | 0.0012 | 439 | 107 | 399 | 14.7 | 396 | 7.2 |
| RZ15 | 95 | 424 | 0.0557 | 0.0023 | 0.5192 | 0.0233 | 0.0670 | 0.0012 | 439 | 94.4 | 425 | 15.6 | 418 | 7.3 |
| RZ16 | 45.2 | 220 | 0.0564 | 0.0027 | 0.4873 | 0.0226 | 0.0622 | 0.0013 | 478 | 103 | 403 | 15.4 | 389 | 8.0 |

**Table 2.** In situ zircon Hf isotopic results of the Luotuoquan granites.

| No | Age (Ma) | 176Yb/177Hf | 176Lu/177Hf | 176Hf/177Hf | 2σ | εHf(0) | εHf(t) | 2σ | TDM1 (Ma) | f(Lu/Hf) | TDM2 (Ma) |
|---|---|---|---|---|---|---|---|---|---|---|---|
| PM12-20: monzogranite | | | | | | | | | | | |
| RZ1 | 409 | 0.014856 | 0.000530 | 0.282242 | 0.000015 | −18.7 | 3.72 | 0.51 | 894 | −0.98 | 1160 |
| RZ2 | 403 | 0.031534 | 0.001173 | 0.28240 | 0.000017 | −13.2 | 2.96 | 0.59 | 927 | −0.96 | 1204 |
| RZ3 | 404 | 0.025478 | 0.000856 | 0.28238 | 0.000018 | −13.9 | 3.82 | 0.64 | 876 | −0.97 | 1150 |
| RZ4 | 405 | 0.041885 | 0.001418 | 0.282381 | 0.000014 | −13.8 | 3.39 | 0.49 | 895 | −0.96 | 1178 |
| RZ5 | 408 | 0.030106 | 0.001031 | 0.282337 | 0.000015 | −15.4 | 3.51 | 0.51 | 887 | −0.97 | 1173 |
| RZ6 | 403 | 0.040490 | 0.001374 | 0.282318 | 0.000015 | −16.1 | 3.71 | 0.54 | 873 | −0.96 | 1156 |
| RZ7 | 404 | 0.048124 | 0.001711 | 0.282350 | 0.000017 | −14.9 | 4.12 | 0.59 | 884 | −0.95 | 1131 |
| RZ8 | 403 | 0.034757 | 0.001174 | 0.282334 | 0.000014 | −15.5 | 2.32 | 0.51 | 934 | −0.96 | 1244 |
| RZ9 | 406 | 0.028390 | 0.000962 | 0.282279 | 0.000014 | −17.4 | 3.12 | 0.51 | 902 | −0.97 | 1196 |
| RZ10 | 403 | 0.032415 | 0.001107 | 0.282360 | 0.000015 | −14.6 | 2.66 | 0.54 | 914 | −0.97 | 1223 |
| RZ11 | 403 | 0.033578 | 0.001125 | 0.282319 | 0.000014 | −16.0 | 3.30 | 0.50 | 889 | −0.97 | 1182 |
| RZ12 | 406 | 0.054288 | 0.001987 | 0.282317 | 0.000014 | −16.1 | 3.88 | 0.49 | 879 | −0.94 | 1148 |
| RZ13 | 402 | 0.008480 | 0.000266 | 0.282205 | 0.000016 | −20.1 | 3.58 | 0.55 | 907 | −0.99 | 1164 |
| RZ14 | 407 | 0.029075 | 0.000976 | 0.282415 | 0.000014 | −12.6 | 4.34 | 0.49 | 873 | −0.97 | 1119 |
| RZ15 | 408 | 0.054699 | 0.001828 | 0.282399 | 0.000015 | −13.2 | 3.74 | 0.54 | 875 | −0.94 | 1158 |
| RZ16 | 407 | 0.029610 | 0.001011 | 0.282348 | 0.000016 | −15.0 | 4.57 | 0.58 | 847 | −0.97 | 1105 |
| PM06-34: syenogranite | | | | | | | | | | | |
| RZ1 | 387 | 0.032029 | 0.001097 | 0.282317 | 0.000018 | −16.1 | 2.96 | 0.64 | 890 | −0.97 | 1192 |
| RZ2 | 392 | 0.011210 | 0.000352 | 0.282254 | 0.000018 | −18.3 | 5.19 | 0.64 | 814 | −0.99 | 1054 |
| RZ3 | 397 | 0.045344 | 0.001502 | 0.282343 | 0.000015 | −15.2 | 2.52 | 0.54 | 914 | −0.95 | 1227 |
| RZ4 | 390 | 0.028869 | 0.000986 | 0.282321 | 0.000019 | −15.9 | 4.66 | 0.67 | 831 | −0.97 | 1086 |
| RZ5 | 398 | 0.031534 | 0.001173 | 0.282400 | 0.000015 | −13.2 | 2.39 | 0.51 | 922 | −0.96 | 1236 |
| RZ6 | 398 | 0.025478 | 0.000856 | 0.282380 | 0.000017 | −13.9 | 2.31 | 0.59 | 928 | −0.97 | 1241 |
| RZ7 | 417 | 0.041885 | 0.001418 | 0.282381 | 0.000016 | −13.8 | 0.95 | 0.58 | 998 | −0.96 | 1341 |
| RZ8 | 388 | 0.030106 | 0.001031 | 0.282337 | 0.000014 | −15.4 | 2.92 | 0.51 | 893 | −0.97 | 1195 |
| RZ9 | 412 | 0.040490 | 0.001374 | 0.282318 | 0.000014 | −16.1 | 1.81 | 0.51 | 958 | −0.96 | 1283 |
| RZ10 | 410 | 0.048124 | 0.001711 | 0.282350 | 0.000014 | −14.9 | 0.90 | 0.50 | 993 | −0.95 | 1339 |
| RZ11 | 419 | 0.034757 | 0.001174 | 0.282334 | 0.000014 | −15.5 | 4.30 | 0.49 | 862 | −0.96 | 1131 |
| RZ12 | 394 | 0.028390 | 0.000962 | 0.282279 | 0.000014 | −17.4 | 2.11 | 0.50 | 927 | −0.97 | 1250 |
| RZ13 | 412 | 0.032415 | 0.001107 | 0.282360 | 0.000013 | −14.6 | 2.98 | 0.46 | 910 | −0.97 | 1209 |
| RZ14 | 396 | 0.033578 | 0.001125 | 0.282319 | 0.000013 | −16.0 | 4.68 | 0.47 | 833 | −0.97 | 1090 |
| RZ15 | 418 | 0.054288 | 0.001987 | 0.282317 | 0.000017 | −16.1 | 2.91 | 0.61 | 922 | −0.94 | 1218 |
| RZ16 | 389 | 0.008480 | 0.000266 | 0.282205 | 0.000016 | −20.1 | 3.12 | 0.58 | 884 | −0.99 | 1183 |

**Table 3.** Major and trace element compositions of the Luotuoquan granites.

| Samples | Monzogranirte | | | | | Syenogranite | | |
|---|---|---|---|---|---|---|---|---|
| | PM03-2 | PM03-26 | PM03-41 | PM12-20 | PM17-4 | D5195 | D8373 | PM06-34 |
| $SiO_2$ | 71.04 | 71.80 | 74.95 | 72.80 | 74.10 | 71.60 | 76.00 | 72.30 |
| $TiO_2$ | 0.29 | 0.32 | 0.14 | 0.29 | 0.12 | 0.49 | 0.18 | 0.17 |
| $Al_2O_3$ | 14.20 | 14.30 | 13.20 | 13.60 | 14.10 | 13.30 | 12.10 | 12.10 |
| $TFe_2O_3$ | 2.41 | 2.61 | 1.11 | 2.57 | 1.08 | 3.63 | 1.70 | 1.86 |
| MnO | 0.06 | 0.06 | 0.01 | 0.05 | 0.01 | 0.03 | 0.01 | 0.03 |
| MgO | 0.55 | 0.63 | 0.27 | 0.41 | 0.30 | 0.62 | 0.07 | 0.63 |
| CaO | 2.16 | 1.98 | 0.76 | 1.33 | 1.29 | 2.00 | 1.03 | 2.53 |
| $Na_2O$ | 3.45 | 3.49 | 3.03 | 3.10 | 3.51 | 3.01 | 2.46 | 2.69 |
| $K_2O$ | 4.29 | 4.02 | 5.59 | 4.93 | 4.87 | 4.03 | 4.89 | 4.89 |
| $P_2O_5$ | 0.12 | 0.12 | 0.03 | 0.09 | 0.04 | 0.11 | 0.04 | 0.29 |
| LoI | 1.52 | 0.77 | 0.76 | 0.92 | 0.51 | 1.23 | 1.43 | 2.48 |
| Total | 100.07 | 100.07 | 99.85 | 100.03 | 99.99 | 100.06 | 99.93 | 99.99 |
| Rb | 211.0 | 132.0 | 167.0 | 177.0 | 215.0 | 112.0 | 114.0 | 134.0 |
| Ba | 406.0 | 444.0 | 398.0 | 345.0 | 466.0 | 556.0 | 499.0 | 755.0 |
| Th | 28.60 | 11.50 | 11.30 | 20.70 | 16.70 | 29.20 | 21.20 | 24.80 |
| U | 4.81 | 2.41 | 2.23 | 2.95 | 1.76 | 2.10 | 1.30 | 1.90 |
| Ta | 1.10 | 0.76 | 0.90 | 1.08 | 0.80 | 0.81 | 0.40 | 0.72 |
| Nb | 14.00 | 9.53 | 9.56 | 11.50 | 10.10 | 15.10 | 5.13 | 12.40 |
| Sr | 90.50 | 50.10 | 60.40 | 65.80 | 84.50 | 86.00 | 59.00 | 79.00 |
| Zr | 333.0 | 202.0 | 245.0 | 259.0 | 236.0 | 365.0 | 147.0 | 275.0 |
| Hf | 10.90 | 6.98 | 8.40 | 8.90 | 8.01 | 12.50 | 6.10 | 9.50 |
| Y | 55.52 | 57.88 | 58.64 | 43.61 | 44.51 | 47.40 | 43.40 | 37.70 |
| Ga | 19.10 | 18.20 | 14.30 | 17.00 | 20.60 | 25.40 | 21.70 | 17.20 |
| La | 40.28 | 25.93 | 19.75 | 47.56 | 42.23 | 93.60 | 48.60 | 68.20 |
| Ce | 93.30 | 56.65 | 45.19 | 108.9 | 83.71 | 189.0 | 89.60 | 140.0 |
| Pr | 10.97 | 7.51 | 5.87 | 14.30 | 11.20 | 25.60 | 12.90 | 19.20 |
| Nd | 43.77 | 31.14 | 24.06 | 55.80 | 44.47 | 94.90 | 50.90 | 74.00 |
| Sm | 9.62 | 7.69 | 6.25 | 11.46 | 9.12 | 17.20 | 11.50 | 14.10 |
| Eu | 1.13 | 0.96 | 0.86 | 1.37 | 1.00 | 1.22 | 1.18 | 1.20 |
| Gd | 7.84 | 6.18 | 5.11 | 9.07 | 7.45 | 15.10 | 9.35 | 11.50 |
| Tb | 1.61 | 1.44 | 1.33 | 1.62 | 1.42 | 2.11 | 1.53 | 1.80 |
| Dy | 9.84 | 9.66 | 9.57 | 8.49 | 8.51 | 9.62 | 8.13 | 9.20 |
| Ho | 1.87 | 1.89 | 1.89 | 1.50 | 1.53 | 1.64 | 1.50 | 1.70 |
| Er | 4.68 | 4.72 | 4.75 | 3.75 | 3.81 | 4.56 | 4.35 | 4.00 |
| Tm | 0.84 | 0.93 | 0.96 | 0.63 | 0.69 | 0.66 | 0.73 | 0.60 |
| Yb | 4.87 | 5.52 | 5.38 | 3.57 | 4.11 | 3.98 | 4.67 | 3.70 |
| Lu | 0.85 | 0.96 | 0.89 | 0.66 | 0.75 | 0.54 | 0.64 | 0.50 |
| $\sum$REE | 231.5 | 161.2 | 131.9 | 268.7 | 220.0 | 459.7 | 245.6 | 349.7 |
| LREE/HREE | 6.14 | 4.15 | 3.41 | 8.17 | 6.78 | 11.00 | 6.95 | 9.57 |
| δEu | 0.39 | 0.41 | 0.45 | 0.40 | 0.36 | 0.23 | 0.34 | 0.27 |

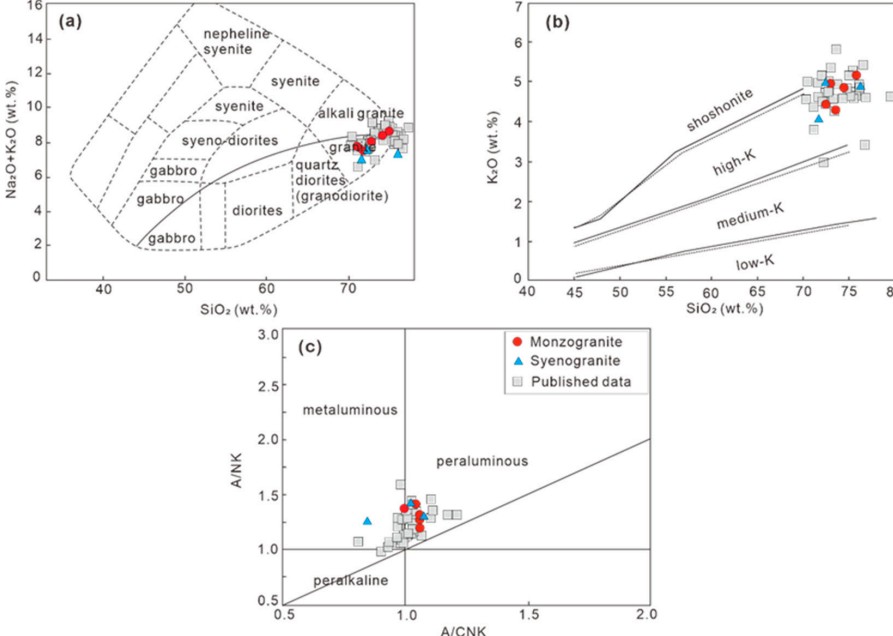

**Figure 5.** (**a**) Total alkalis ($Na_2O + K_2O$) vs. $SiO_2$ diagram [49]; (**b**) $K_2O$ vs. $SiO_2$ diagram [50]; (**c**) A/NK vs. A/CNK diagram [51]. Published data of the early Devonian granitoids in the Beishan area are from [4,6,30,31,42,52,53].

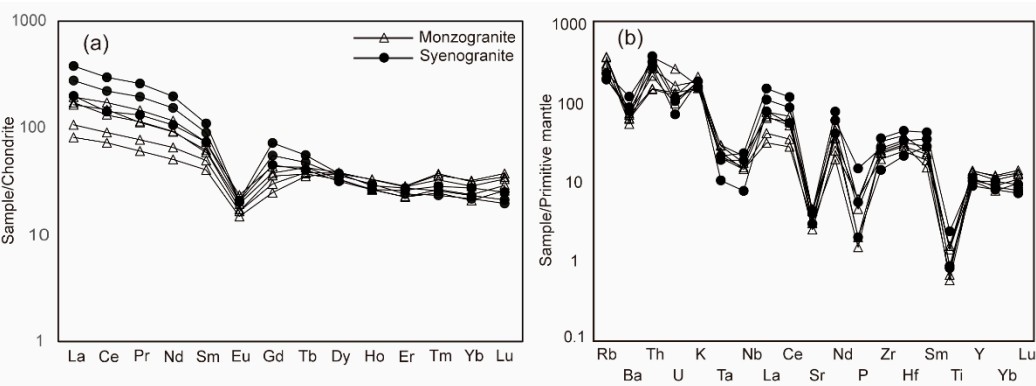

**Figure 6.** Chondrite-normalized REE element patterns (**a**) and primitive mantle-normalized trace element spider diagrams (**b**) for the Luotuoquan granites. Normalization values are from [54].

## 5. Discussion

### 5.1. Devonian Magmatism in the Southern Beishan Orogenic Belt

The LA–ICP–MS zircon U–Pb ages obtained in this study (404–399 Ma) are consistent with the timing of pluton emplacement in the SBOB, including the Liuyuan monzogranite (397 ± 7 Ma) [40], Shuangfengshan A-type granites (415 ± 3 Ma) [4], Huitongshan K-feldspar granite (397 ± 3 Ma) [42], and Shijinpo granitoid (404 ± 2 Ma) [53]. Previous studies have established that the Silurian–Devonian was a crucial period of magmatism in the BOB (Figure 1a), with age peaks of 440, 420, and 400 Ma. The Luotuoquan granites, as products of Devonian magmatism, offer valuable insights into the geodynamic evolution of the BOB during the mid-Paleozoic.

### 5.2. Petrogenesis and Magma Sources

The syenogranite samples in this study have undergone mylonitization, as evidenced by petrographic observation (Figure 2d). Therefore, it is imperative to evaluate the impact of alteration on both major and trace elements before discussing their petrogenesis [55]. Our samples demonstrate differentiated correlations between "immobile alteration" elements (e.g., Zr) and the other trace elements (Figure 7), where Nb, Ta, Th, and Hf display a stronger correlation with Zr than Rb and Ba, suggesting that this alteration has less effect on these elements. Consequently, the subsequent discussion will primarily focus on more immobile elements such as Nb, Ta, Zr, and REEs.

The Luotuoquan monzogranite and syenogranite exhibit similar geochemical characteristics and are discussed together. They have high $SiO_2$ (71.04–76.00 wt.%) and $K_2O + Na_2O$ (7.04–8.62 wt.%) but low $Al_2O_3$ (mean of 13.36 wt.%), MgO (mean of 0.03 wt.%), and CaO (mean of 1.64 wt.%), indicating A-type granite geochemical characteristics. Nevertheless, highly fractionated I-type and S-type granites also share similarities with A-type granites. Highly fractionated S-type granites tend to exhibit higher $P_2O_5$ (mean of 0.14 wt.%) and lower $Na_2O$ (mean of 2.81 wt.%) than A-type granites [56]. The monzogranite–syenogranite displays low $P_2O_5$ (mean of 0.11 wt.%) and high $Na_2O$ (mean of 3.09 wt.%), indicating it does not belong to the highly fractionated S-type granite. Compared with highly fractionated I-type granite, A-type granite shows iron enrichment and magnesium depletion with higher $Fe_2O_3^T/MgO$ ratios. The $Fe_2O_3^T/MgO$ value of Luotuoquan granite ranges from 2.95 to 24.28 (mean of 6.95), which is significantly greater than that of highly fractionated I-type granite (2.27) and S-type granite (2.38) [57,58]. Additionally, typical A-type granites are enriched in trace elements such as Th, Nb, Ta, Zr, Hf, Ga, and Y while being depleted in Sr, Ti, P, Cr, Co, Ni, V, etc., with obvious negative Eu anomaly [57,59]. All samples exhibit high Th, Zr, K, Ga, Y, and Yb but low Sr, P, Eu, and Ti, which also implies that Luotuoquan granite had A-type geochemical features. Previous studies suggested that A-type granitoids share similar geochemistry characteristics of high $10,000 \times Ga/Al$ values (>2.6) and Zr + Nb + Ce + Y (>350 ppm) [57]. The discrimination diagrams (Figure 8)

demonstrate that the majority, if not all, of the samples are plotted within the A-type granite field. We, therefore, conclude that the Luotuoquan monzogranite–syenogranite pluton are typical A-type intrusions.

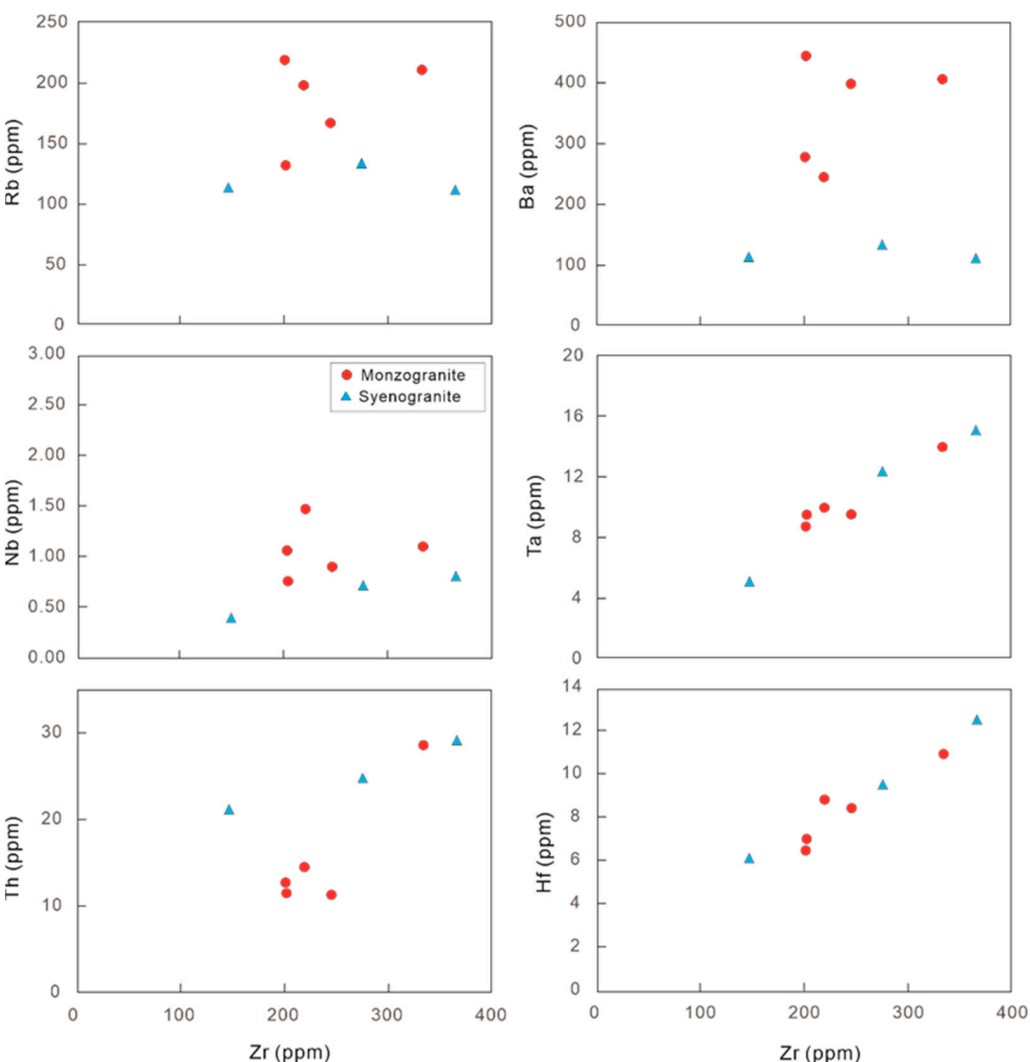

**Figure 7.** Trace elements vs. Zr for Luotuoquan A-type granites.

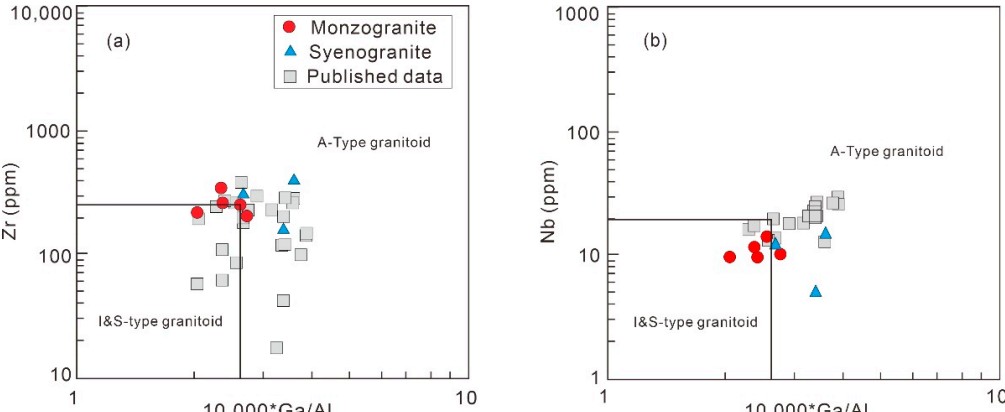

**Figure 8.** *Cont.*

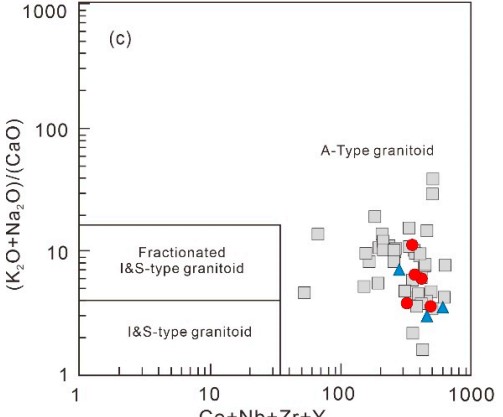
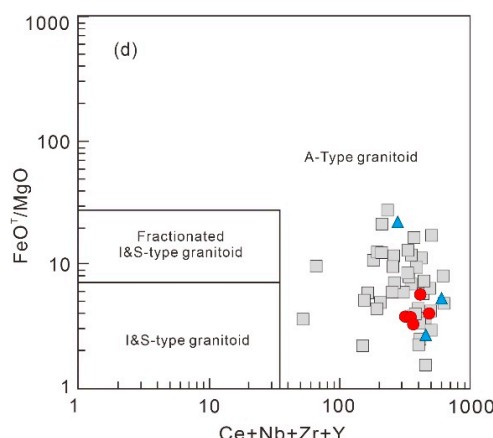

**Figure 8.** Discrimination diagrams for early Devonian granitoids from BOB. Zr (**a**) and Nb (**b**) vs. $10,000 * Ga/Al$ and $(K_2O + Na_2O)/CaO$ (**c**) and $FeO^T/MgO$ (**d**) vs. $Ce + Nb + Zr + Y$ diagrams [57]. Published data are from the same references as in Figure 5.

Various petrogenetic models have been proposed for A-type granites: (1) high crystallization differentiation of mantle-derived basaltic magma [60–63]; (2) mixing of mantle-derived and crustal materials [64–67]; (3) Partial melting of lower crustal material: partial melting of granulite facies remnants after granitic magma extraction [55,56,59,68], and partial melting of calc-alkaline tonalite-granodiorite [69]; (4) the melting of lower crustal rocks due to heating by mantle magma underplating [58,70,71]. Mafic rocks contemporaneous with A-type granite are rare in the study area, and no mafic microgranular enclaves are found in the granites, indicating little crust–mantle mixing. Experimental petrology shows residual granulite facies in the lower crust are low in K, Si and high in Ca, Al, and Mg [69], which cannot explain the production of Luotuoquan A-type granite rich in Si, alkali, low in Al and Mg by partial melting.

The $CaO/Na_2O$ ratios distinguish between pelite-derived melts ($CaO/Na_2O < 0.5$) and melts derived from greywackes or igneous sources ($CaO/Na_2O = 0.3–1.5$) [72]. High-temperature melts generally exhibit lower $Al_2O_3/TiO_2$ ratios than low-temperature melts [73]. The monzogranite–syenogranite is characterized by high $CaO/Na_2O$ ratios (mean of 0.53) and low $Al_2O_3/TiO_2$ ratios (mean of 63.73), suggesting a source of metagreywackes or metamorphic igneous rock. The samples are all plotted within the field of partial melts derived from metagreywackes in the molar $Al_2O_3/(MgO + FeO^T)$ vs. molar $CaO/(MgO + FeO^T)$ and molar $K_2O/Na_2O$ vs. molar $CaO/(MgO + FeO^T)$ diagrams (Figure 9). Therefore, the granitic magma source is likely related to partial melts from metagreywackes.

The Hf isotope analysis of U-Pb dated zircon grains can trace the original magma sources and distinguish between the reworking of continental crust and the remelting of juvenile crust [74,75]. In this study, the early Devonian Luotuoquan monzogranite–syenogranite has considerably positive zircon $\varepsilon_{Hf}(t)$ of +0.9–+5.2 (Figure 10) and slightly young two-stage Hf model ages of 1.05–1.34 Ga (Figure 10). The SBOB is thought to have developed abundant Mesoproterozoic to Neoproterozoic basement complex, which is a suite of metamorphosed clastic rock [17,18,20,37]. The significantly positive $\varepsilon_{Hf}(t)$ values and relatively young two-stage Hf model ages suggest that the granitic rocks originated from either the depleted mantle or through partial melting of recently accreted juvenile crustal material within the depleted mantle. Relevant mafic rocks are rarely contemporaneous, and the absence of mafic microgranular enclaves suggests limited direct involvement of newly derived mantle magma in granite formation. Therefore, the early Devonian granitoids from the SBOB primarily originated from partial melting of the overlying Mesoproterozoic crust facilitated by the underplating of mantle-derived magma.

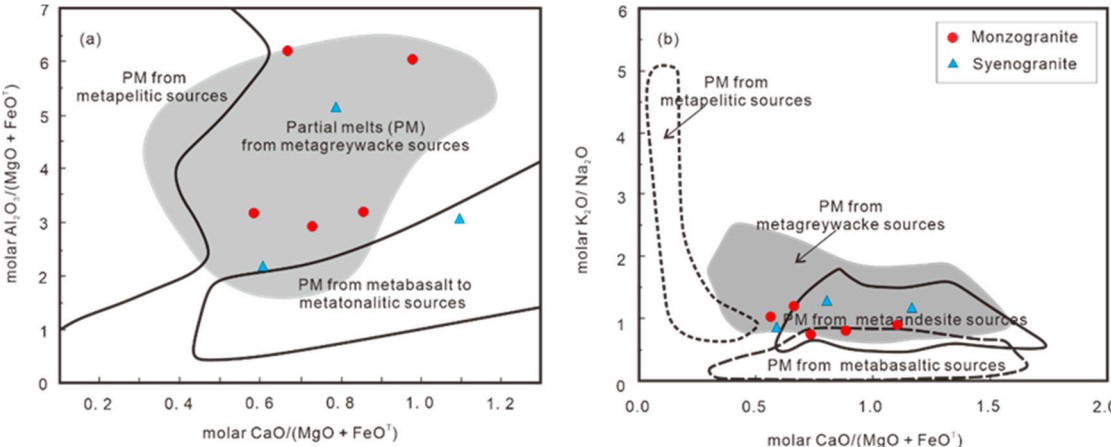

**Figure 9.** Discrimination diagrams of the source area of the Luotuoquan monzogranite and syenogranite. (**a**) molar $Al_2O_3/(MgO + FeO^T)$ vs. molar $CaO/(MgO + FeO^T)$ [76]; (**b**) molar $K_2O/Na_2O$ vs. molar $CaO/(MgO + FeO^T)$ [77].

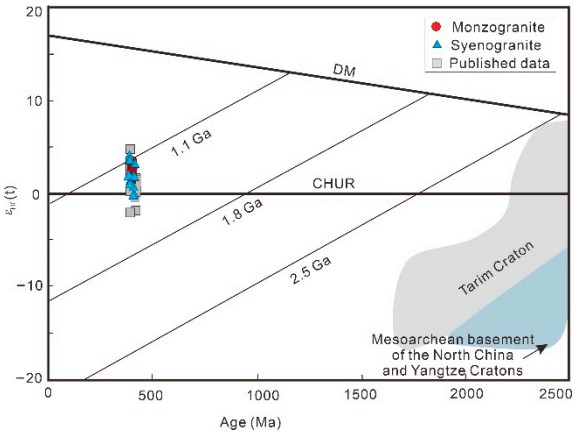

**Figure 10.** Zircon $\varepsilon_{Hf}(t)$ vs. Age (Ma) diagrams of granitoids from the SBOB (base map after [75]). Published data of granitoids from [6,42]. The data on Tarim, North China, and Yangtze blocks were from [78].

*5.3. Geodynamic Evolution*

Recent research suggests that the BOB is an accretionary orogen with multiple island arcs and mélange belts that experienced subduction–collision during the Paleozoic [5,12, 28,30,37]. The zircon U–Pb age of the ophiolite complexes in the central part of the BOB is 536–519 Ma [7,79–81], and the lower Cambrian Shuangyingshan Formation unconformable overlies the Neoproterozoic Xichangjing Formation [38], indicating the Beishan Ocean was formed before the Cambrian. Previous studies have proposed that the magmatic effects of slab windows vary dramatically during ridge subduction, forming adakitic magmas due to slab melting and A-type magmas resulting from asthenosphere upwelling through the slab window [54,82]. Widespread early Paleozoic magmatic rocks (464–424 Ma) [6,10,16, 32,37,40,83,84] and ophiolitic mélange (462–420 Ma) [9,13,39,85] in the BOB indicate slab subduction setting during the Ordovician to Devonian.

The A-type granite has garnered significant attention due to its distinctive tectonic background. The Luotuoquan A-type granites fall within the volcanic arc and within-plate granites on tectonic discrimination diagrams (Figure 11a,b) rather than pertaining to ocean ridge granites. The studied granites are generally plotted within the A2 field (Figure 11c,d). The A2-type granitoids represent magmas sourced from the underplated crust or continental crust that has experienced a cycle of island-arc magmatism or continent-continent collision [86]. We propose that their potential origins were associated with

island-arc magmatism. The monzogranite in the Hongliuhe ophiolite has a weighted age of 412.4 ± 2.9 Ma, indicating that the closure of the oceanic basin was completed prior to this event [41]. The granites from the Liuyuan area, with ages ranging from 436-423 Ma [40], 415 Ma [42], and 397 Ma [4,40], represent post-collision background products that are possibly associated with subduction plate detachment. Moreover, recent discoveries have revealed the presence of early Devonian post-collision granites in the middle of BOB. U–Pb dating has determined that these granites range in age from 402 to 387 Ma [15,31,35,52]. The aforementioned evidence suggests that the formation of Luotuoquan A-type granites can be attributed to a post-collision tectonic setting.

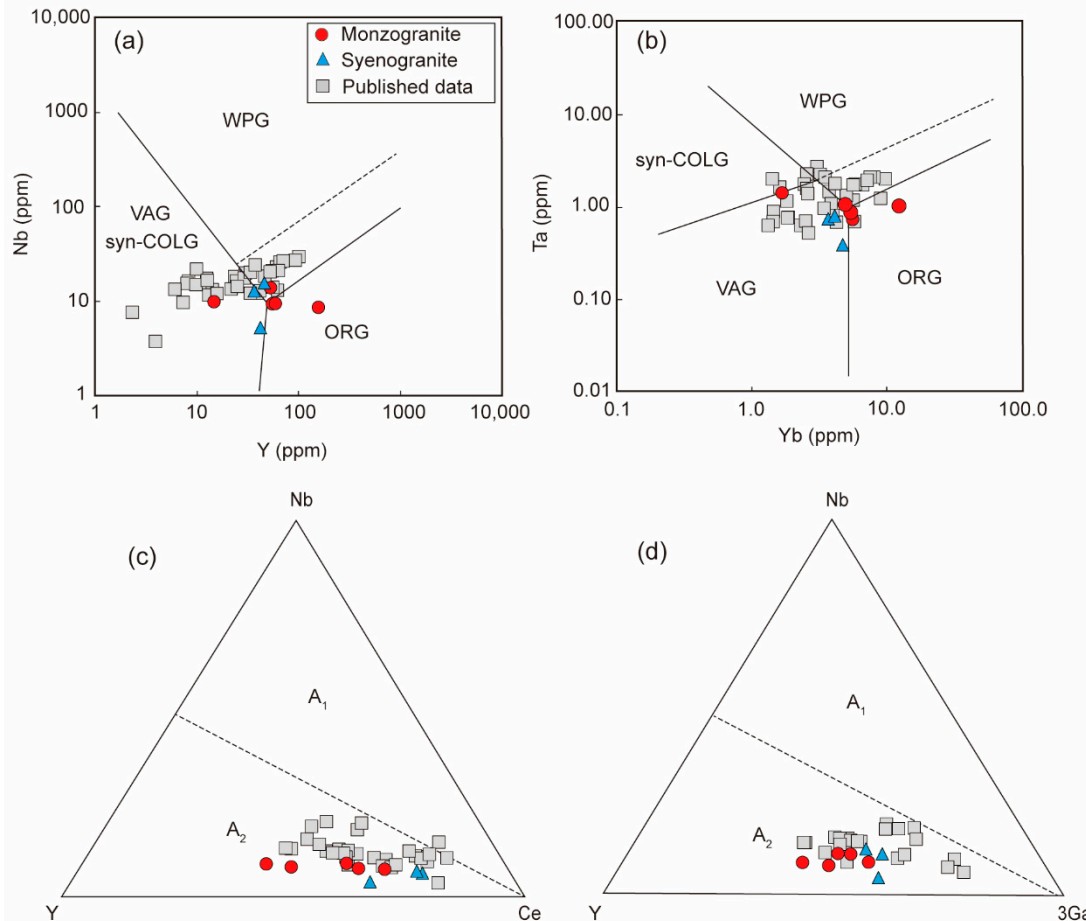

**Figure 11.** Nb vs. Y (**a**), Ta vs. Yb (**b**), Nb-Y-Ce (**c**), Nb-Y-3Ga (**d**) tectonic discriminant diagrams of the early Devonian granitoids from BOB ((**a**,**b**) modified after [56], (**c**,**d**) modified after [86]). Syn-COLG-syn-collision granites; VAG-volcanic arc granites; WPG-within plate granites; ORG-ocean ridge granites; post-COLG-post collision granites. Published data are from the same references as in Figure 5.

Whole-rock geochemical data of 460–390 Ma granites from the SBOB were collected to calculate zircon saturation temperatures ($\ln D_{Zr} = (10{,}108 \pm 32)/T(K) - (1.16 \pm 0.15)(M - 1) - (1.48 \pm 0.09))$ [87], $M = (Na + K + 2Ca)/(Al \times Si)(mol))$ [88] and crustal thickness ($H = [Sr/Y + (42.03 \pm 6.28)]/(1.49 \pm 0.15))$ [89,90]. During the early Paleozoic, a significant increase in crustal growth occurred due to the melting of subducting oceanic crust, resulting in the emplacement of large amounts of granitoids in the SBOB from 452 to 424 Ma [6]. This process was accompanied by a gradual decrease in zircon saturation temperatures and an increase in crustal thickness (Figure 11). The A-type granites, which were formed between 415 and 397 Ma [4,42], exhibit a higher Zr saturation temperature range of 755–831 °C and a thinner crust thickness ranging from 32 to 28 km (Figure 12). These findings suggest that the SBOB had already transitioned into an extensional setting during the early Devonian.

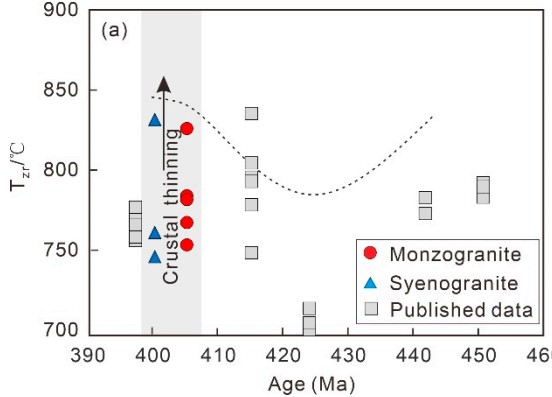 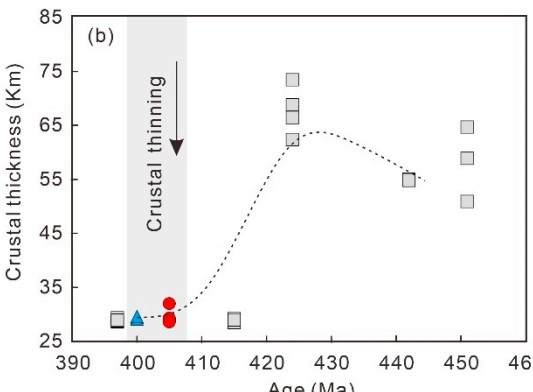

**Figure 12.** $T_{zr}$ vs. Age (**a**) and Crustal thickness vs. Age (**b**) plots for Paleozoic granitoids in the SBOB. Published data from [4,6,42].

In summary, the Luotuoquan A-type granites yield U-Pb ages of approximately 404–399 Ma, indicating post-collision extensional setting during the early Devonian. This evidence indicates a post-collision extension setting in the early Devonian and implies the closure of the Beishan Ocean prior to this time.

## 6. Conclusions

(1) The zircon U–Pb ages of the Luotuoquan monzogranite and syenogranite are 399.4 ± 5.1 Ma and 404.9 ± 1.4 Ma, respectively.

(2) The petrographic and geochemical signatures of the Luotuoquan monzogranite and syenogranite indicate they are A-type granites and were emplaced in a post-collision extensional setting. Furthermore, these granites are the result of partial melting primarily from Mesoproterozoic crusts composed mainly of metagreywackes.

(3) The occurrence of early Devonian granitoids suggests that SBOB had already undergone extensional tectonics following the closure of the Beishan Ocean during this period.

**Author Contributions:** Conceptualization: W.C. and M.Z.; Data curation: Q.Y., X.F. and J.Z.; Investigation: W.C., G.T. and X.L.; Methodology: M.Z. and G.T.; Resources: W.C. and M.Z.; Writing–original draft preparation: W.C., M.Z. and G.T.; Writing–review and editing: W.C. and M.Z.; Visualization: Q.Y. and X.F.; Supervision: X.L.; Project administration: M.Z. and Q.Y. All authors have read and agreed to the published version of the manuscript.

**Funding:** This research was supported by the Natural Science Foundation of Hebei Province (D2020403044) and the Opening Foundation of Hebei Key Laboratory of Strategic Critical Mineral Resources (HGU-SCMR2304).

**Data Availability Statement:** All the data are presented in the paper.

**Acknowledgments:** We sincerely appreciate the editors and reviewers for their constructive comments, which helped to significantly improve the manuscript.

**Conflicts of Interest:** The authors declare no conflict of interest.

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
