# Peer review of "Ocean–Continent Conversion in Beishan Orogenic Belt: Evidence from Geochemical and Zircon U-Pb-Hf Isotopic Data of Luotuoquan A-Type Granite"

_minerals, doi:10.3390/min13111411_

Round 1
Reviewer 1 Report
Comments and Suggestions for Authors
This is an interesting manuscript, which discusses the origin and tectonic significance of the Luotuoquan granites from the Central Asian Orogenic belt. It presents whole-rock major and trace element analyses as well as zircon U-Pb-Hf isotopic data. My main criticism is the small amount of analytical data. It includes eight whole rock analyses and zircon analyses from two samples. I think it is a very limited database, although the data from the literature partially compensate for this shortcoming.
Additional comments:
1. Line 127 the authors should give information about analytical precision and accuracy so that readers would not have to search for this essential info in the literature. In addition, I am not sure if the reference cited [43] is the correct one. It refers to in situ analyses of anhydrous minerals - not to whole-rock analyses.
2. Line 211-213. According to Rudnick and Gao (2003) the average of the continental crust of Zr/Hf ratio is ~ 36, virtually the same as that of the mantle. Considering the analytical error, the values of crust and mantle are about the same. The ratio cannot be used to distinguish between mantle and crust sources.
3. Line 237 Al2O3 – numbers should be subscripts
4. Figure 8 needs corrections “metapelitic”, “metatonalitic”, “metabasaltic”
5. According to petrography (page 3), the rocks are gneissic and exhibit a mylonitic texture. Could these processes modify the chemical composition, particularly mobile elements such as Rb? If so could figures 10c and 10d be suspect? Comments on the possible effect of alteration on chemical composition should be included.
6. Line 268 Figure 10 is not very convincing that the rocks are within plate. More likely they are post-orogenic. See Pearce 1996 [84].
7. Zircon saturation temperatures. Watson and Harrison calculations were modified by Boehnke et al. 2013. Chemical Geology 351, 324-334. You should use the recent procedure.
Comments on the Quality of English Language
Minor editing of English required
Author Response
Dear Editors and reviewer:
Thank you for your letter and for the reviewers’ comments concerning our manuscript entitled “Ocean-continent conversion in Beishan Orogenic Belt: Evidence from geochemical and zircon U-Pb-Hf isotopic data of Luotuoquan A-type granite” (ID: minerals-2541741). Those comments are all valuable and very helpful for revising and improving our paper, as well as the important guiding significance to our research. We have studied comments carefully and have made correction. Revised portions are marked in yellow in the paper. The main corrections in the paper and the responds to the reviewer’s comments are as following.
Comments to the Author
This is an interesting manuscript, which discusses the origin and tectonic significance of the Luotuoquan granites from the Central Asian Orogenic belt. It presents whole-rock major and trace element analyses as well as zircon U-Pb-Hf isotopic data. My main criticism is the small amount of analytical data. It includes eight whole rock analyses and zircon analyses from two samples. I think it is a very limited database, although the data from the literature partially compensate for this shortcoming.
Additional comments:
- Line 127 the authors should give information about analytical precision and accuracy so that readers would not have to search for this essential info in the literature. In addition, I am not sure if the reference cited [43] is the correct one. It refers to in situ analyses of anhydrous minerals - not to whole-rock analyses.
Response: We have updated “3.3 Whole-rock geochemical analysis” according to the Reviewer’s suggestion (see Line 112-117).
- Line 211-213. According to Rudnick and Gao (2003) the average of the continental crust of Zr/Hf ratio is ~ 36, virtually the same as that of the mantle. Considering the analytical error, the values of crust and mantle are about the same. The ratio cannot be used to distinguish between mantle and crust sources.
Response: This part of the discussion was removed based on the Reviewer’s suggestion.
- Line 237 Al2O3 – numbers should be subscripts
Response: We are very sorry for our negligence. We have modified according to the Reviewer’s suggestion (see Line 224).
- Figure 8 needs corrections“metapelitic”, “metatonalitic”, “metabasaltic”
Response: We are very sorry for our negligence. We have modified Figure 8 according to the Reviewer’s suggestion (see Line 222).
- According to petrography (page 3), the rocks are gneissic and exhibit a mylonitic texture. Could these processes modify the chemical composition, particularly mobile elements such as Rb? If so could figures 10c and 10d be suspect? Comments on the possible effect of alteration on chemical composition should be included.
Response: The granite samples in this study have undergone mylonitization, as evidenced by petrographic observation (Figure 2d). Therefore, it is imperative to evaluate the impact of alteration on both major and trace elements before discussing their petrogenesis. Our samples demonstrate good correlations between "immobile alteration" elements (e.g., Zr) and the majority of other elements (e.g., Nb, La, Th, Hf) (Figure 7), suggesting that the influence of such alteration on most elements is negligible. Consequently, the subsequent discussion will exclude elements influenced by alteration (e.g., Rb, Ba), instead focusing on more immobile elements such as Sr, REEs, and HFSEs (see line 168-174).
- Line 268 Figure 10 is not very convincing that the rocks are within plate. More likely they are post-orogenic. See Pearce 1996 [84].
Response: The relevant geochemical and geochronological data are consistent with a post-collision extension setting in the early Devonia. Geodynamic evolution has been discussed in more detail (see line 236-249 and 267-275).
- Zircon saturation temperatures. Watson and Harrison calculations were modified by Boehnke et al. 2013. Chemical Geology 351, 324-334. You should use the recent procedure.
Response: We have modified according to the Reviewer’s suggestion (see Line 38-43). Whole-rock geochemical data of 460–390 Ma granites from the SBOB were collected to calculate zircon saturation temperatures (ln DZr = (10108 ± 32)/T(K) – (1.16 ± 0.15) (M – 1) – (1.48 ± 0.09), M = (Na + K + 2Ca)/(Al × Si)(mol)) and crustal thickness (H = [Sr/Y + (42.03 ± 6.28)]/(1.49 ± 0.15) (see line). A-type granites 415–397 Ma have formed with a higher Zr saturation temperature (755℃–831℃) and thinner crust thickness (32–28 km) (see line 256-257 and line 263).
Other changes:
5.2 Petrogenesis and Magma sources has been discussed in more detail (see line).
We tried our best to improve the manuscript and made some changes in the manuscript. These changes will not influence the content and framework of the paper.
We appreciate for Editors/Reviewers’ warm work earnestly, and hope that the correction will meet with approval.
Once again, thank you very much for your comments and suggestions.
Thank you and best regards.
Yours sincerely,
Corresponding author:
Name: Minjie Zhang
E-mail: zmj0794@163.com

Reviewer 2 Report
Comments and Suggestions for Authors
Chen et al. have investigated granites from northern China employing geochemical methods to date them (using U-Pb in zircon) and constrain their origin and geotectonic context. The conclusions that they (i) represent A-type granites and (ii) formed, about 400 Ma ago, in a post-collisional extensional setting are permissible but not entirely convincing, judging from their classification diagrams where the granites mostly occupy a boundary region. Hf isotopic compositions of the zircons are compatible with their source rocks being derived from depleted mantle during the Mesoproterozoic.
The manuscript is well written and the topic certainly of interest for a wide readership in eastern Asia.
There are several points that I would like to see addressed in a revised version:
1.) The rocks are distinguished into monzogranites and syenogranites with very different modal abundances of plagioclase and K-feldspar, despite very similar bulk-rock compositions for SiO2, K2O and total alkalis. How is this possible?
2.) U-Pd zircon dating of one monzogranite yields a weighted age of 404.6±1.4 Ma whereas zircons from a syenogranite provide an age of 399.4±5.1 Ma, that is their ages overlap within error. Nevertheless, the authors state that the syenogranite is the younger of the two and refer to Fig. 2. However I have failed to see definite proof for this inference. Moreover, I wonder why the young granite shows weak mylonitization when the older granite does not.
3.) Section 3.3 provides very little information about the analytical methods including precision and accuracy of the trace element analyses. I point out that Y/Ho ratios of the granites are surprisingly variable. The same is true for Th/U. In addition, half of the samples show negative "Er anomalies". I therefore would like to see some information on the accuracy of their data.
Also, one sample with a peculiar REE pattern (the only rock with relative light REE depletion) was omitted from Fig. 6 without giving a reason.

Author Response
Dear Editors and reviewer:
Thank you for your letter and for the reviewers’ comments concerning our manuscript entitled “Ocean-continent conversion in Beishan Orogenic Belt: Evidence from geochemical and zircon U-Pb-Hf isotopic data of Luotuoquan A-type granite” (ID: minerals-2541741). Those comments are all valuable and very helpful for revising and improving our paper, as well as the important guiding significance to our research. We have studied comments carefully and have made correction. Revised portions are marked in yellow in the paper. The main corrections in the paper and the responds to the reviewer’s comments are as following.
Comments to the Author
Chen et al. have investigated granites from northern China employing geochemical methods to date them (using U-Pb in zircon) and constrain their origin and geotectonic context. The conclusions that they (i) represent A-type granites and (ii) formed, about 400 Ma ago, in a post-collisional extensional setting are permissible but not entirely convincing, judging from their classification diagrams where the granites mostly occupy a boundary region. Hf isotopic compositions of the zircons are compatible with their source rocks being derived from depleted mantle during the Mesoproterozoic.
The manuscript is well written and the topic certainly of interest for a wide readership in eastern Asia.
There are several points that I would like to see addressed in a revised version:
1.) The rocks are distinguished into monzogranites and syenogranites with very different modal abundances of plagioclase and K-feldspar, despite very similar bulk-rock compositions for SiO2, K2O and total alkalis. How is this possible?
Response: It is identified as monzogranites and syenogranites according to petrography. The Devonian monzogranites and syenogranites from Beishan area show similar bulk-rock compositions are consistent with the previous (Bai et al., 2020; Chen et al., 2020; Pan et al., 2019; Wang et al., 2009; Zhao et al.,2007; Zheng et al., 2012).
2.) U-Pd zircon dating of one monzogranite yields a weighted age of 404.6±1.4 Ma whereas zircons from a syenogranite provide an age of 399.4±5.1 Ma, that is their ages overlap within error. Nevertheless, the authors state that the syenogranite is the younger of the two and refer to Fig. 2. However I have failed to see definite proof for this inference. Moreover, I wonder why the young granite shows weak mylonitization when the older granite does not.
Response: The monzogranite mass is characterized by the trend of gradually decreasing grain size from the center to the edges, the syenogranite is in the center of the study area and intruded into monzogranite due to field investigation. The monzogranite and syenogranite were sampled in different locations, and mylonitization was developed in syenogranite due to the influence of NW shear zone at Wengeshan area.
3.) Section 3.3 provides very little information about the analytical methods including precision and accuracy of the trace element analyses. I point out that Y/Ho ratios of the granites are surprisingly variable. The same is true for Th/U. In addition, half of the samples show negative "Er anomalies". I therefore would like to see some information on the accuracy of their data.
Also, one sample with a peculiar REE pattern (the only rock with relative light REE depletion) was omitted from Fig. 6 without giving a reason.
Response: The section on "3.3 Whole-rock geochemical analysis" has been revised in accordance with the suggestions made by the Reviewer (see Line 80-82). I checked and modified table 3 carefully and kept two valid digits. Once again, please accept my apologies for this mistake.
Other changes:
5.2 Petrogenesis and Magma sources has been discussed in more detail (see line).
5.3 Geodynamic evolution has been discussed in more detail (see line).
We tried our best to improve the manuscript and made some changes in the manuscript. These changes will not influence the content and framework of the paper.
We appreciate for Editors/Reviewers’ warm work earnestly, and hope that the correction will meet with approval.
Once again, thank you very much for your comments and suggestions.
Thank you and best regards.
Yours sincerely,
Corresponding author:
Name: Minjie Zhang
E-mail: zmj0794@163.com

Reviewer 3 Report
Comments and Suggestions for Authors
This paper applies U-Pb dating, elements and Hf isotopes to constrain characteristics of Luotuoquan A-type granite. The analytical results are authentic and most of the discussions are reasonable. However, there still remain some improper contents in the manuscript. Especially ocean-continent conversion in the title of the article is obviously not reflected in the discussion section. It is suggested to add this part or modify the title. In addition, pay attention to spelling mistakes. The charts must be checked carefully. A moderate revision is suggested for its publication in Minerals.
Specific comments:
Replace all figures with all high definition pictures.
Keep two valid digits in the table.
Table 1 Should be “isotopic ratios”.
Table 2 and 3: Notice the superscript and subscript.
Figure 4: The legend loses the explanation of the yellow circle.
Line 65 Shibanshan Unite? It should be Shuangyingshan Unit in the Fig. 1b.
Line 19-22: This part is biased from the discussion part (Line 228-234).
Although there is a deviation in the figure 5, syenogranite does not show more alkali-rich characteristics than monzogranite. Please give the basis for the classification and naming of your granite.
5.2. Petrogenesis and Magma sources section:
Fig 7 shows that monzogranite is not typical A-type granite, similar to highly fractionated I-type granite. (Saijun Sun 2015 Lithos; Junjie Zhang 2023 Chemical Geology)
Other evidence is needed for the source area of A-type granite, such as trace elements.
Line 228-234: The source region of metagraywacks to explain the positive Hf isotope combines and the thin crust is unconvincing.
5.3. Geodynamic evolution section:
Please replace figure 10 with the latest Pearce diagram.
It is indisputable that the A-type granite is in the extensional background, but whether it is caused by subduction or collision, the author needs to demonstrate in detail. Please look at Zhang`s paper (Junjie Zhang 2023 Chemical Geology).
Comments on the Quality of English LanguagePay attention to spelling mistakes
Author Response
Dear Editors and reviewer:
Thank you for your letter and for the reviewers’ comments concerning our manuscript entitled “Ocean-continent conversion in Beishan Orogenic Belt: Evidence from geochemical and zircon U-Pb-Hf isotopic data of Luotuoquan A-type granite” (ID: minerals-2541741). Those comments are all valuable and very helpful for revising and improving our paper, as well as the important guiding significance to our research. We have studied comments carefully and have made correction. Revised portions are marked in yellow in the paper. The main corrections in the paper and the responds to the reviewer’s comments are as following.
Comments to the Author:
This paper applies U-Pb dating, elements and Hf isotopes to constrain characteristics of Luotuoquan A-type granite. The analytical results are authentic and most of the discussions are reasonable. However, there still remain some improper contents in the manuscript. Especially ocean-continent conversion in the title of the article is obviously not reflected in the discussion section. It is suggested to add this part or modify the title. In addition, pay attention to spelling mistakes. The charts must be checked carefully. A moderate revision is suggested for its publication in Minerals.
Specific comments:
Replace all figures with all high definition pictures.
Response: We are very sorry for our negligence. We have modified all pictures according to the Reviewer’s suggestion.
Keep two valid digits in the table.
Response: We have modified all data in tables according to the Reviewer’s suggestion.
Table 1 Should be “isotopic ratios”.
Response: We are very sorry for our negligence. We have modified according to the Reviewer’s suggestion (see Line 124).
Table 2 and 3: Notice the superscript and subscript.
Response: It is a great pity that all superscript and subscript changed after the manuscript was submitted. Anyway, the superscript and subscript were re-checked carefully.
Figure 4: The legend loses the explanation of the yellow circle.
Response: We are very sorry for our negligence. We have modified the legend of Figure 4 according to the Reviewer’s suggestion (see Line 135-136).
Line 65 Shibanshan Unite? It should be Shuangyingshan Unit in the Fig. 1b.
Response: We are very sorry for our negligence. The SBOB is situated within the Hongliuhe and Liuyuan ophiolitic mélanges and comprises the Shuangyingshan and Huaniushan Units (see Line 60).
Line 19-22: This part is biased from the discussion part (Line 228-234).
Response: We are very sorry for our negligence. their tDM2 values fall between 1.05 and 1.34 Ga (See line 21, 134, 218).
Although there is a deviation in the figure 5, syenogranite does not show more alkali-rich characteristics than monzogranite. Please give the basis for the classification and naming of your granite.
Response: It is identified as monzogranites and syenogranites according to petrography (see Line 74-77). The Devonian monzogranites and syenogranites from Beishan area show similar bulk-rock compositions are consistent with the previous (Bai et al., 2020; Chen et al., 2020; Pan et al., 2019; Wang et al., 2009; Zhao et al.,2007; Zheng et al., 2012).
5.2. Petrogenesis and Magma sources section:
Fig 7 shows that monzogranite is not typical A-type granite, similar to highly fractionated I-type granite. (Saijun Sun 2015 Lithos; Junjie Zhang 2023 Chemical Geology)
Other evidence is needed for the source area of A-type granite, such as trace elements.
Response: We have modified Petrogenesis and Magma sources section according to the Reviewer’s suggestion. The addition of further evidence (Fe2O3T/MgO value and trace element analysis) and more comprehensive discussion has been added into section 5.2 (see Line 182-193).
Line 228-234: The source region of metagraywacks to explain the positive Hf isotope combines and the thin crust is unconvincing.
Response: The Luotuoquan granite (404-399 Ma) in this paper exhibits the characteristics of A-type granite, which is formed through partial remelting of crustal materials and is associated with lithospheric mantle extension and crustal thinning during this period.
5.3. Geodynamic evolution section:
Please replace figure 10 with the latest Pearce diagram.
Response: Our samples demonstrate good correlations between "immobile alteration" elements (e.g., Zr) and the majority of other elements (e.g., Nb, La, Th, Hf) (Figure 7), suggesting that the influence of such alteration on most elements is negligible. Consequently, the subsequent discussion will exclude elements influenced by alteration (e.g., Rb, Ba), instead focusing on more immobile elements such as Sr, REEs, and HFSEs. Geodynamic evolution has been discussed in more detail (see line 168-174). We have modified Figure 10 and 5.3 Geodynamic evolution (see line 236-253).
It is indisputable that the A-type granite is in the extensional background, but whether it is caused by subduction or collision, the author needs to demonstrate in detail. Please look at Zhang`s paper (Junjie Zhang 2023 Chemical Geology).
Response: t is suggested that the collision between the Dunhuang and Southern Beishan blocks may occur during the late Silurian to early Devonian, which was closely related to the closure of the western segment of the Central Asian Ocean, along with the orogenic event and crustal thickening. The Devonian Luotuoquan granite was generated from the partial remelting of a crustal source, which was probably formed by underplating of mantle-derived magmas of a delamination, relating to the partial separation of the subducted plate, and the extension of the lithospheric mantle during around 400 Ma (see 5.3 Geodynamic evolution; line 236-249 and 267-275).
We tried our best to improve the manuscript and made some changes in the manuscript. These changes will not influence the content and framework of the paper.
We appreciate for Editors/Reviewers’ warm work earnestly, and hope that the correction will meet with approval.
Once again, thank you very much for your comments and suggestions.
Thank you and best regards.
Yours sincerely,
Corresponding author:
Name: Minjie Zhang
E-mail: zmj0794@163.com

Round 2
Reviewer 1 Report
Comments and Suggestions for Authors
This is an improved version of the manuscript. However, I still have some minor points:
1. Lines 173-174 high field strength elements are Th, Zr an Hf while large ion lithophile element are Ba. Sr and Eu
2. Line 212 What is “T”, probably it should read Ti
3. Line 215 discriminated should read “discrimination”
4. Line 470 reference is not complete- the second author is missing
Comments on the Quality of English LanguageMinor editing of English language required
Author Response
Dear reviewer:
Thank you for your comments concerning our manuscript entitled “Ocean-continent conversion in Beishan Orogenic Belt: Evidence from geochemical and zircon U-Pb-Hf isotopic data of Luotuoquan A-type granite” (ID: minerals-2541741). Those comments are all valuable and very helpful for revising and improving our paper, as well as the important guiding significance to our research. We have studied comments carefully and have made correction. Revised portions are marked in yellow in the paper. The main corrections in the paper and the responds to the reviewer’s comments are as following.
Comments to the Author
This is an improved version of the manuscript. However, I still have some minor points:
- Lines 173-174 high field strength elements are Th, Zr an Hf while large ion lithophile element are Ba. Sr and Eu
Response: We are very sorry for our negligence. We have modified according to the reviewer’s suggestion (see Line 153-155).
- Line 212 What is “T”, probably it should read Ti
Response: We are very sorry for our negligence. We have modified according to the reviewer’s suggestion (see Line 189).
- Line 215 discriminated should read “discrimination”
Response: We are very sorry for our negligence. We have modified according to the reviewer’s suggestion (see Line 191).
- Line 470 reference is not complete- the second author is missing
Response: We are very sorry for our negligence. We have checked and modified all references according to the reviewer’s suggestion (see line 299-300, 363, 368-369, 376, 393, 457, and 528).
We tried our best to improve the manuscript and made some changes in the manuscript. These changes will not influence the content and framework of the paper.
We appreciate for reviewer’s warm work earnestly, and hope that the correction will meet with approval.
Once again, thank you very much for your comments and suggestions.
Thank you and best regards.
Yours sincerely,
Corresponding author:
Name: Minjie Zhang
E-mail: zmj0794@163.com